# Molecular determinants in Frizzled, Reck, and Wnt7a for ligand-specific signaling in neurovascular development

**Chris Cho[1], Yanshu Wang[1,2], Philip M Smallwood[1,2], John Williams[1,2], Jeremy Nathans[1,2,3,4]\***

[1]Department of Molecular Biology and Genetics, Johns Hopkins University School of Medicine, Baltimore, United States; [2]Howard Hughes Medical Institute, Johns Hopkins University School of Medicine, Baltimore, United States; [3]Department of Neuroscience, Johns Hopkins University School of Medicine, Baltimore, United States; [4]Department of Ophthalmology, Johns Hopkins University School of Medicine, Baltimore, United States

**Abstract** The molecular basis of Wnt-Frizzled specificity is a central question in developmental biology. Reck, a multi-domain and multi-functional glycosylphosphatidylinositol-anchored protein, specifically enhances beta-catenin signaling by Wnt7a and Wnt7b in cooperation with the 7-transmembrane protein Gpr124. Among amino acids that distinguish Wnt7a and Wnt7b from other Wnts, two clusters are essential for signaling in a Reck- and Gpr124-dependent manner. Both clusters are far from the site of Frizzled binding: one resides at the amino terminus and the second resides in a protruding loop. Within Reck, the fourth of five tandem repeats of an unusual domain with six-cysteines (the CC domain) is essential for Wnt7a stimulation: substitutions P256A and W261A in CC4 eliminate this activity without changing protein abundance or surface localization. Mouse embryos carrying $Reck^{P256A,W261A}$ have severe defects in forebrain angiogenesis, providing the strongest evidence to date that Reck promotes CNS angiogenesis by specifically stimulating Wnt7a and Wnt7b signaling.
DOI: https://doi.org/10.7554/eLife.47300.001

**\*For correspondence:**
jnathans@jhmi.edu

## Introduction

Vascularization of the brain and retina requires beta-catenin (i.e. canonical Wnt) signaling in vascular endothelial cells (ECs) (*Xu et al., 2004*; *Stenman et al., 2008*; *Daneman et al., 2009*). Later in development and in the mature CNS, beta-catenin signaling is required to generate and maintain the blood-brain barrier (BBB) and its retinal counterpart, the blood-retina barrier (BRB) (*Liebner et al., 2008*; *Stenman et al., 2008*; *Daneman et al., 2009*). In both of these contexts, ligands Wnt7a, Wnt7b, and/or Norrin are produced by CNS glia and/or neurons to activate Frizzled (Fz) receptors and Lrp5/6 co-receptors on ECs (reviewed in *Sun and Smith, 2018*; *Wang et al., 2019*). In humans, mutations in the genes coding for beta-catenin, Norrin, Fz4, Lrp5, and the Norrin-specific co-activator Tspan12 cause inherited defects in retinal vascularization (*Sun and Smith, 2018*; *Wang et al., 2019*). Targeted mutations in the corresponding murine genes, as well as in the genes coding for Wnt7a and Wnt7b, cause defects in retinal and/or brain angiogenesis and barrier formation (*Sun and Smith, 2018*; *Wang et al., 2019*).

Mammalian genomes code for 19 Wnts and 10 Frizzleds, and the genomes of other vertebrate classes code for similarly large numbers of Wnts and Frizzleds. A long-standing question in this field is how Wnts and Frizzleds interact so that signaling occurs via the appropriate subset of ligand-receptor pairs. Characterization of Wnt-Frizzled binding in vitro and the effects of different

combinations of Wnts and Frizzleds on beta-catenin signaling in transfected cells, together with analyses of genetic redundancy among Wnt and Frizzled genes, suggest that there is both specificity and promiscuity in Wnt-Frizzled interactions (*Bhat, 1998*; *Bhanot et al., 1999*; *Hsieh et al., 1999*; *Mulroy et al., 2002*; *Stenman et al., 2008*; *Yu et al., 2010*; *Ye et al., 2011*; *Yu et al., 2012*; *Dijksterhuis et al., 2015*; *Voloshanenko et al., 2017*). The question of Wnt-Frizzled specificity has been brought into sharper focus by the three-dimensional co-crystal structure of *Xenopus* Wnt8 and the ligand-binding cysteine-rich domain (CRD) of murine Fz8 (*Janda et al., 2012*). In this structure, XWnt8 resembles a hand that uses only the thumb and one finger to contact the CRD. Much of the contact surface on the amino-terminal lobe of XWnt8 (the 'thumb') is contributed by a covalently attached lipid that is common to all Wnts, while much of the contact surface on the carboxy-terminal lobe (the 'finger') is contributed by evolutionarily conserved amino acids. Thus, these two contact interfaces likely account for only part of the biological specificity of Wnt-Frizzled binding.

A partial answer to the specificity question is emerging from the study of Wnt7a and Wnt7b signaling in the context of CNS angiogenesis and BBB maintenance. Two membrane proteins that are expressed in CNS ECs – the seven-transmembrane protein Gpr124 and the multi-domain glycosyl-phosphatidylinositol (GPI)-anchored protein Reck – specifically enhance signaling via Wnt7a and Wnt7b (*Zhou and Nathans, 2014*; *Posokhova et al., 2015*; *Vanhollebeke et al., 2015*; *Ulrich et al., 2016*; *Cho et al., 2017*; *Eubelen et al., 2018*). Prenatally, EC-specific mutation of Gpr124 or Reck severely impairs CNS angiogenesis (*Kuhnert et al., 2010*; *Anderson et al., 2011*; *Cullen et al., 2011*; *Zhou and Nathans, 2014*; *Cho et al., 2017*). Additionally, postnatal elimination of Reck and Gpr124, together with loss of Norrin, compromises BBB integrity (*Zhou and Nathans, 2014*; *Cho et al., 2017*; *Wang et al., 2018*). Recent biochemical studies of the interactions between Wnt7a/7b, Frizzled, Gpr124, and Reck have begun to dissect the domains and individual amino acids required for their function and for the exquisite ligand specificity that Gpr124 and Reck impart (*Posokhova et al., 2015*; *Cho et al., 2017*; *Eubelen et al., 2018*; *Vallon et al., 2018*). The present study adds to this body of work by (i) comparing the roles of different Fz CRD and transmembrane domains in Wnt7a/Fz/Gpr124/Reck signaling, and (ii) defining amino acids in Wnt7a that are required for Gpr124- and Reck-dependence, and (iii) defining amino acids in Reck that are required for Wnt7a-dependent signaling and complex formation. In mice, CRISPR/Cas9-mediated alanine substitutions at two critical amino acids in Reck cause a severe defect in CNS angiogenesis and likely represents a clean elimination of Wnt7a/7b stimulation without affecting the structure or function of other Reck domains.

## Results

### Frizzled CRD specificity in Reck-Gpr124-Wnt7a signaling and complex formation

Among the ten members of the Frizzled family, Fz5, Fz8, and to a lesser extent Fz4 facilitate Reck-Gpr124-Wnt7a signaling and ligand/receptor/co-activator association on the surface of transfected cells, whereas Fz3 and Fz6 do not (*Vanhollebeke et al., 2015*; *Cho et al., 2017*). To explore the Frizzled domain(s) responsible for this specificity, we examined binding of Reck domains CC1-5 fused to alkaline phosphatase (AP) to intact (live) cells displaying Gpr124, Wnt7a, and various full length Fz proteins or GPI-anchored Fz CRDs. [The N-terminal region of Reck contains five tandem copies of an ~60 amino acid domain with a characteristic pattern of six cysteines (*Takahashi et al., 1998*); we refer to these as CC domains.] With full-length Fz targets, Reck(CC1-5)-AP binds Fz5 = Fz8>>Fz4>Fz6, a pattern that is closely matched by the corresponding FzCRDs displayed as Myc-tagged and GPI anchored targets (*Figure 1A*; summarized in *Figure 1C*). Specifically, Fz4, Fz5, Fz6, and Fz8 CRD-Myc-GPI proteins accumulate at the cell surface of living cells to approximately the same level, and they bind Wnt7a-1D4 with comparable efficiencies — as shown, respectively, by anti-Myc and anti-1D4 binding to intact cells — but only Fz5 and Fz8 CRDs confer high levels of Reck(CC1-5)-AP binding (*Figure 1B*). Thus, the Reck(CC1-5)-AP binding signals reflect a specificity for particular Fz CRD sequences rather than differences in the abundances of cell-surface Wnt7a-1D4 or FzCRD-Myc-GPI.

Since a subset of Fz CRDs are permissive for Reck-Gpr124-Wnt7a complex formation, we next asked whether weakly and strongly permissive CRDs (Fz4 and Fz8, respectively) promote Reck(CC1-

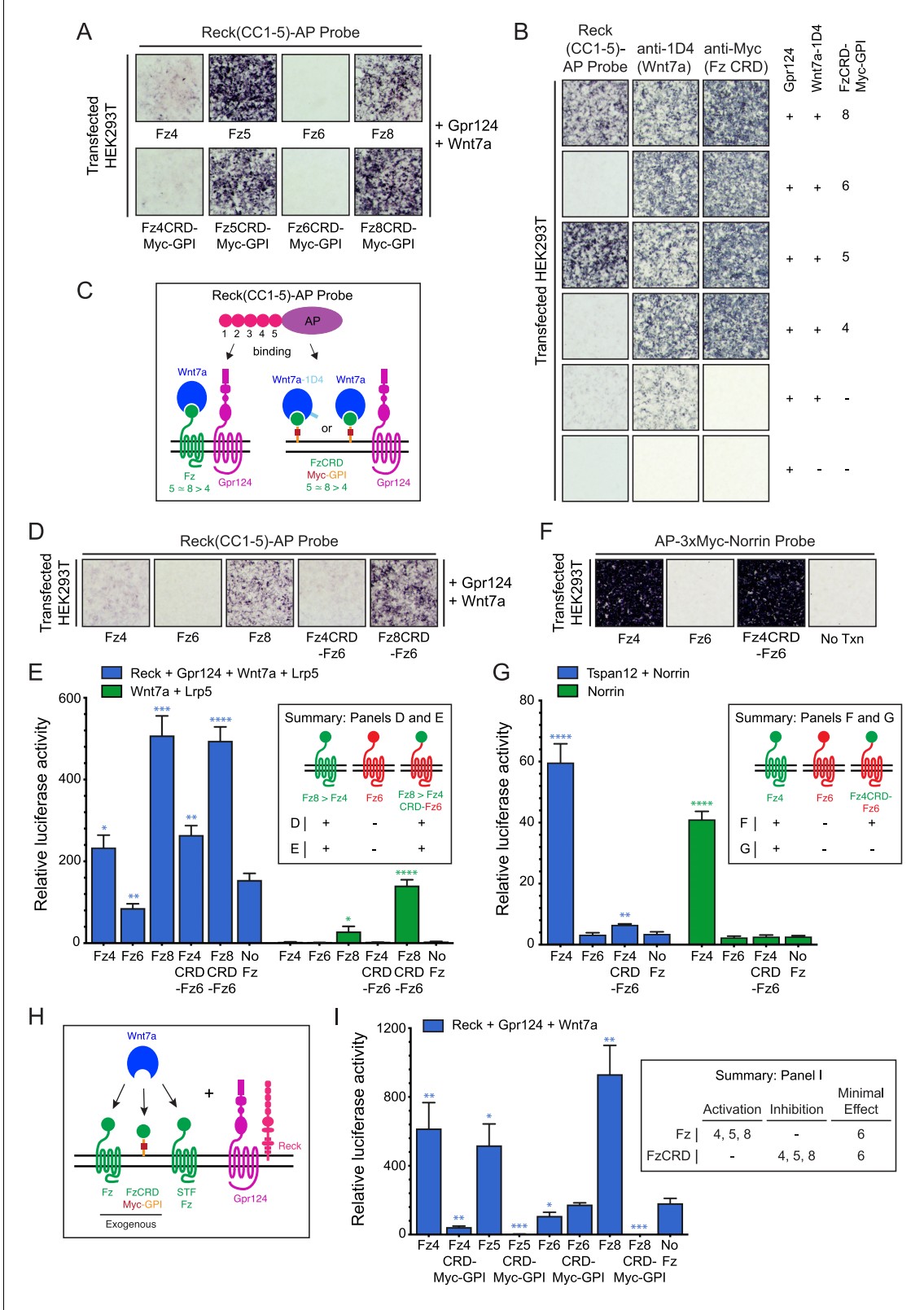

**Figure 1.** Frizzled CRD specificity for Reck-Gpr124-Wnt7a binding and signaling. (**A**) Reck(CC1-5)-AP binding to live HEK293T cells transfected with Gpr124, Wnt7a, and full-length Fz (top) or Gpr124, Wnt7a, and FzCRD-Myc-GPI (bottom). (**B**) Reck(CC1-5)-AP binding as in (**A**), with Gpr124, Wnt7a, and the indicated FzCRD-Myc-GPI targets, together with anti-Myc and anti-1D4 controls. (**C**) Summary of the AP binding assay in (**A**) and (**B**). (**D**) Reck(CC1-5)-AP binding as in (**A**), with WT or chimeric Frizzleds. (**E**) Beta-catenin signaling assay using STF cells transfected with Wnt7a, Gpr124, Reck, and Lrp5

*Figure 1 continued on next page*

*Figure 1 continued*

(left) or Wnt7a and Lrp5 (right), together with WT or chimeric Frizzleds. Inset: summary of AP binding (**D**) and STF signaling (**E**). In this and subsequent figures, bars represent mean ± SD. Statistical significance, determined by the unpaired t-test, is represented by * (p<0.05), ** (p<0.01), *** (p<0.001), and **** (p<0.0001). The statistical comparisons in (**E**), (**G**), and (**I**) are to the 'No Fz' control. (**F**) AP-3xMyc-Norrin binding assay as in (**A**), with WT or chimeric Frizzleds. (**G**) Beta-catenin signaling assay using STF cells transfected with Tspan12 and Norrin (left) or Norrin (right), together with WT or chimeric Frizzleds. Inset: summary of AP binding (**F**) and STF signaling (**G**). (**H**) Schematic of the FzCRD-Myc-GPI competition experiment in (**I**). (**I**) The effect of FzCRD-Myc-GPI competition on beta-catenin signaling by Reck, Gpr124, and Wnt7a. Inset: summary of STF signaling.

DOI: https://doi.org/10.7554/eLife.47300.002
The following source data is available for figure 1:

**Source data 1.** Relative luciferase activity for STF experiments in *Figure 1* panels E, G, and I.
DOI: https://doi.org/10.7554/eLife.47300.003

5)-AP binding if they are fused to the 7-transmembrane region of a non-permissive Fz (Fz6). Upon co-transfection with Gpr124 and Wnt7a, we observed weak binding of Reck(CC1-5)-AP to Fz4CRD-Fz6 (a hybrid with the Fz4CRD joined to the Fz6 linker and transmembrane domains) and strong binding of Reck(CC1-5)-AP to Fz8CRD-Fz6 (*Figure 1D*), signal intensities that mirror those of full-length Fz4 and Fz8 (*Figure 1A*). In a luciferase reporter cell line for beta-catenin signaling [Super TOP Flash (STF) cells; *Xu et al., 2004*, co-transfection with Reck, Gpr124, Wnt7a, Lrp5, and WT or chimeric Frizzleds showed that signaling by Fz4CRD-Fz6 and Fz8CRD-Fz6 was comparable to WT Fz4 and Fz8, respectively (*Figure 1E*; *Figure 1—source data 1*). Fz6 showed no STF activity compared to the no-Fz control. Wnt7a signaling by both WT and chimeric Fz4 and Fz8 proteins was enhanced by Reck and Gpr124 (*Figure 1E*; compare left and right halves; *Figure 1—source data 1*). Taken together, these data show that part of the specificity for Wnt7a-Fz-Reck-Gpr124 complex formation and signaling resides within the Fz CRD.

As a point of comparison, we conducted an analogous experiment with Norrin and Fz4, which form a high affinity ligand-receptor complex that activates beta-catenin signaling in conjunction with the co-activator Tspan12 (*Xu et al., 2004*; *Junge et al., 2009*). In contrast to the results with Wnt7a, while AP-3xMyc-Norrin binds with comparable efficiencies to HEK293T cells expressing Fz4 or Fz4CRD-Fz6, Norrin-dependent beta-catenin signaling in STF cells was dramatically enhanced with Fz4 but was minimally above background with Fz4CRD-Fz6 (*Figure 1F and G*; *Figure 1—source data 1*). These data are consistent with the independent observation that the linker region between the Fz4 CRD and transmembrane domain enhances Norrin binding and signaling (*Bang et al., 2018*).

As an independent measure of Frizzled CRD specificity in the context of Wnt7a signaling, we quantified the inhibition of Wnt7a/Fz/Reck/Gpr124 signaling upon co-transfection with different FzCRD-Myc-GPI proteins (*Figure 1H and I*; *Figure 1—source data 1*). [In STF cells, low-level expression of multiple Fz genes likely accounts for beta-catenin signaling in the absence of a co-transfected Fz (*Zhou and Nathans, 2014*; *Eubelen et al., 2018*) and therefore co-expression of FzCRD-Myc-GPI might be expected to compete for the ligand and other signaling components (*Figure 1H*).] As seen in *Figure 1I*, beta-catenin signaling induced by Wnt7a, Reck, and Gpr124 was completely inhibited by Fz5CRD-Myc-GPI and Fz8-CRD-Myc-GPI; it was strongly inhibited by Fz4CRD-Myc-GPI; and it was unaffected by Fz6CRD-Myc-GPI (*Figure 1—source data 1*). This rank order of inhibition matches the rank order of cell-surface complex formation (*Figure 1A, B and D*), lending further support to the conclusion that part of the specificity in Wnt7a-Reck-Gpr124 signaling resides within the Fz CRD. The inhibition rather than activation of signaling by FzCRD-Myc-GPI proteins implies an essential role for the Frizzled 7-TM domain in signaling.

## Molecular determinants for Wnt7a function in Reck- and Gpr124-mediated signaling

Wnt7a and Wnt7b are the only Wnts that exhibit Reck- and Gpr124-stimulated signaling (*Zhou and Nathans, 2014*; *Posokhova et al., 2015*; *Vanhollebeke et al., 2015*; *Cho et al., 2017*). As a first step in defining the region(s) of Wnt7a that account for this specificity, we generated chimeras between Wnt7a and either Wnt3 or Wnt3a, with fusion points at the junction between the N- and C-terminal domains of the horseshoe-shaped Wnt (*Janda et al., 2012*; *Hirai et al., 2019*;

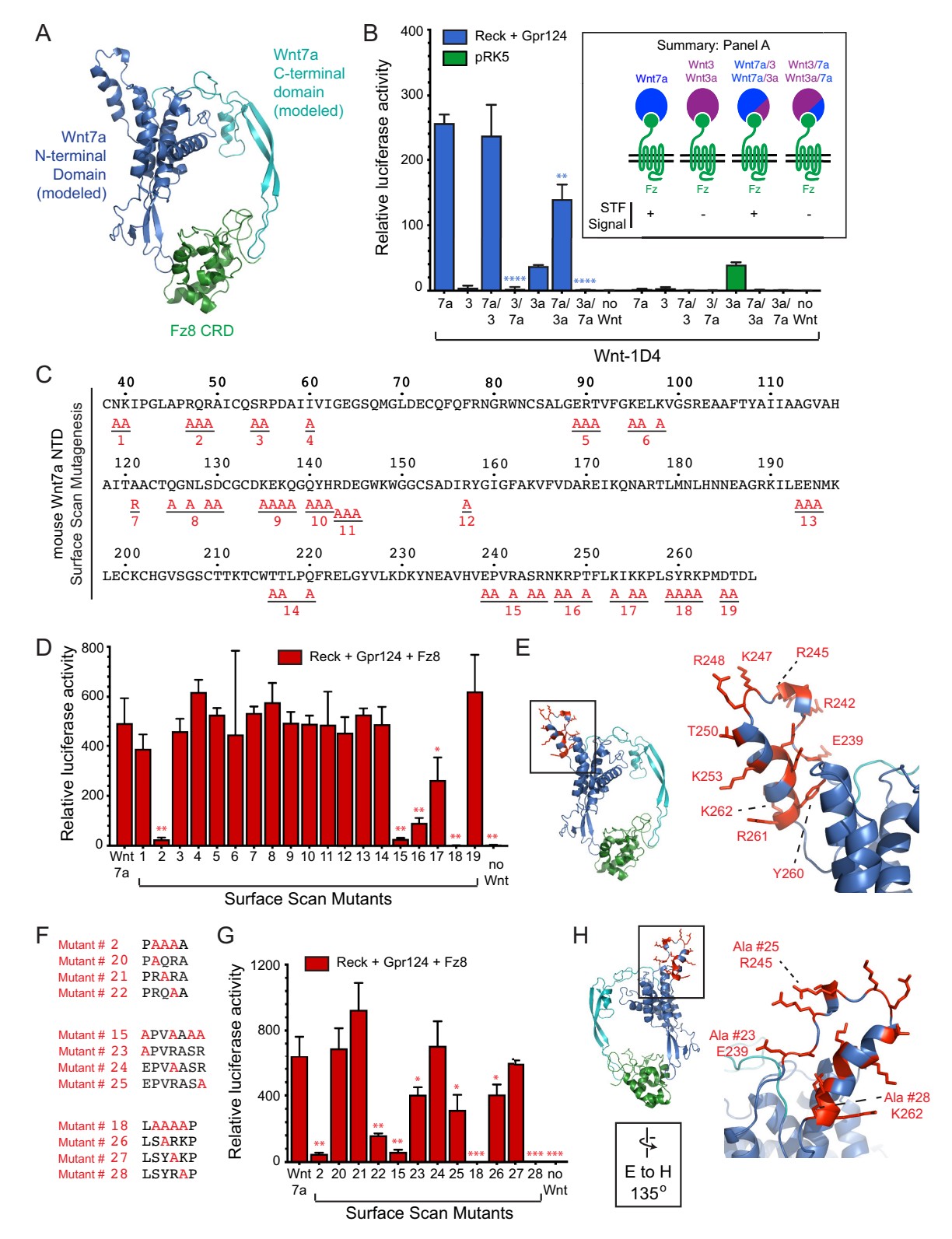

**Figure 2.** Wnt7a regions that are required for Reck/Gpr124-stimulated signaling. (**A**) Backbone model of Wnt7a (N-terminal domain, blue; C-terminal domain, cyan) bound to Fz8 CRD (green) based on the Wnt-CRD crystal structure of *Janda et al. (2012)*. The N-terminal domain of Wnt7a consists of ~270 amino acids (some of which were not resolved in the crystal structure), and the C-terminal domain consists of ~80 amino acids. (**B**) Beta-catenin signaling assay using STF cells transfected with Reck and Gpr124 (left) or pRK5 vector control (right), together with the indicated Wnts. Inset: summary

*Figure 2 continued on next page*

*Figure 2 continued*

of STF signaling. Statistical comparisons in (B), (D), and (G) are to WT Wnt7a. (C) Amino acid sequence of the N-terminal domain of mouse Wnt7a, with alanine scanning mutants indicated. (D) Beta-catenin signaling assay using STF cells transfected with Reck, Gpr124, and Fz8, together with WT or mutant Wnt7a. (E) Left, backbone model of Wnt7a bound to Fz8 CRD as in (A), with amino acids that are critical for Wnt7a signaling shown in red. Right, the boxed region is displayed at higher magnification. (F) Single alanine substitution mutants of Wnt7a, indicated in red. (G) Beta-catenin signaling assay as in (D) with WT or the indicated Wnt7a mutants. (H) Left, backbone model of Wnt7a bound to Fz8 CRD as in (A) except rotated 135 degrees, with amino acids that are critical for Wnt7a signaling shown in red. Right, the boxed region is displayed at higher magnification.

DOI: https://doi.org/10.7554/eLife.47300.004

The following source data and figure supplements are available for figure 2:

**Source data 1.** Relative luciferase activity for STF experiments in *Figure 2* panels B, D, and G.
DOI: https://doi.org/10.7554/eLife.47300.008
**Figure supplement 1.** Production of intact and chimeric Wnt proteins for Reck- and Gpr124-mediated signaling.
DOI: https://doi.org/10.7554/eLife.47300.005
**Figure supplement 2.** Locations of alanine mutations on the Wnt7a-Fz8CRD model.
DOI: https://doi.org/10.7554/eLife.47300.006
**Figure supplement 3.** Cell-surface CRD binding by Wnt7a alanine mutants.
DOI: https://doi.org/10.7554/eLife.47300.007

*Figure 2A*). As an adjunct to these experiments, we deployed the live-cell immunostaining assay shown in *Figure 1B* to provide a semi-quantitative estimate of the level of expression of correctly folded Wnts by measuring the accumulation of C-terminally 1D4 epitope-tagged Wnt bound to a GPI-anchored CRD at the cell-surface. This assay has the advantage that it imposes two criteria for Wnt immunostaining: (1) transit through the ER-to-plasma membrane quality control system, and (2) Fz CRD binding. Wnt surface localization does not appear to reflect non-specific sticking because it is substantially increased by co-expression of FzCRD-myc-GPI (*Figure 1B*). The low level of surface Wnt seen in the absence of a co-expressed FzCRD-myc-GPI likely reflects binding to endogenous full-length Frizzleds (*Figure 1B*). As seen in *Figure 2—figure supplement 1*, with serial 2-fold reductions of transfected Wnt7a-1D4 plasmid, there is a corresponding monotonic reduction in (i) the cell surface accumulation of Wnt7a-1D4 in the presence of Fz8CRD-Myc-GPI (*Figure 2—figure supplement 1A*) and (ii) the abundance of Wnt7a in whole cell lysates, as assessed by immunoblotting (*Figure 2—figure supplement 1B*). These data imply that cell surface immunostaining provides a semi-quantitative estimate of the abundance of correctly folded and CRD bound Wnt. Using this method, we found that the parental and chimeric Wnts were produced at comparable levels (*Figure 2—figure supplement 1C and D*). Reck- and Gpr124-stimulated signaling in STF cells was only observed when the N-terminal domain (NTD) was derived from Wnt7a (i.e., Wnt7a, Wnt7a/3, and Wnt7a/3a; *Figure 2B*; *Figure 2—source data 1*).

To further refine the molecular determinants for Reck/Gpr124 stimulation of Wnt7a signaling, groups of surface-exposed residues in the NTD of Wnt7a were mutated to alanine. By modeling Wnt7a based on the XWnt8-mFz8CRD structure, we defined a set of surface-exposed residues within the NTD that are highly conserved between Wnt7a and Wnt7b but are not shared with other Wnts. These Wnt7a/7b-specific surface residues were interrogated with a set of 19 alanine-scanning mutants, each with a C-terminal 1D4 epitope tag (*Figure 2C*; *Figure 2—figure supplement 2*). All of the Wnt mutants were expressed at levels similar to WT Wnt7a, as determined by cell-surface immuno-staining following co-transfection with Fz8CRD-Myc-GPI (*Figure 2—figure supplement 3*). Upon transfection of STF cells with Reck, Gpr124, and Fz8, together with WT or mutant Wnt7a, we observe complete loss of function for mutant #18, a near-complete loss of function for mutants #2 and #15, and a partial loss of function for mutants #16 and #17 (*Figure 2D*; *Figure 2—source data 1*). While the amino acid sequence encompassed by mutant #2 is in a region that is disordered in the XWnt8-mFz8CRD structure, mutants #15-#18 correspond to a protruding and highly charged region far from the mFz8 CRD binding site (*Figure 2E*).

For the mutants with the largest defects (#2, #15, and #18), individual residues were changed to alanine to identify the ones that are critical for Wnt7a activity (*Figure 2F*). Mutants #20, #21, #24, and #27 had little or no effect on signaling; mutants #22, #23, #25, and #26 partially reduced signaling; and mutant #28 (K262A) completely abolished signaling (*Figure 2F–H*; *Figure 2—source data 1*). All of the Wnt7a mutants were expressed at levels similar to WT Wnt7a (*Figure 2—figure*

*supplement 3*). These data define a small number of amino acids within two distinct regions of Wnt7a that play an important role in Reck- and Gp124-stimulated signaling and are unlikely to interact with the Fz CRD.

## The Reck CC4 domain is critical for multi-protein complex formation and Wnt7a signaling

Using a series of Reck CC domain deletion mutants, we previously reported that CC1 and CC4 play important roles in Wnt7a signaling in STF cells and that CC1 interacts with Gpr124 (*Cho et al., 2017*). To further explore the role(s) of the CC domains, we probed the surface of live HEK293T cells transfected with Wnt7a, Fz5, and Gpr124 with AP fusions to CC1, CC1-2, CC1-3, CC1-4, or CC1-5 (*Figure 3A and B*). Binding was observed only for Reck(CC1-4)-AP and Reck(CC1-5)-AP, and the strength of the binding signal was greatly reduced in the absence of over-expressed Gpr124. These data indicate that CC4 is essential for multi-protein complex formation and CC5 is dispensable, consistent with our earlier observation that deletion of CC4 eliminated Reck-dependent stimulation of Wnt7a beta-catenin signaling in STF cells, while deletion of CC5 had little or no effect (*Cho et al., 2017*).

To test whether the Reck CC1 and CC4 domains were sufficient for Gpr124-Fz-Wnt7a signaling and multi-protein complex formation, we generated deletion constructs that replaced CC2 and CC3 with either a short or a long glycine/serine spacer in the context of full-length Reck and as a Reck (CC1-5, ΔCC2-3)-AP fusion. In the context of full-length Reck, the two deletion mutants ('Reck ΔCC2-3') accumulated on the cell surface at levels roughly comparable to WT Reck (*Figure 3—figure supplement 1A*). In STF cells, the two Reck ΔCC2-3 mutants activated Wnt7a signaling 4- to 6-fold less efficiently than WT Reck (*Figure 3—figure supplement 1B*; *Figure 3—source data 1*). Interestingly, the Reck(CC1-5, ΔCC2-3)-AP probes bound to cells transfected with Wnt7a, Fz5, and Gpr124 at levels similar to that of a WT probe (*Figure 3—figure supplement 1C*; summarized in *Figure 3—figure supplement 1D*). These data imply that Reck CC1 and CC4 are sufficient to facilitate Wnt7a- and Gpr124-dependent multi-protein complex assembly, but full signaling activity likely depends on an appropriate spacing between the CC1 and CC4 domains.

To identify the key residues responsible for Reck CC4 function, groups of evolutionarily conserved and/or polar/charged CC4 residues were mutated to alanine (*Figure 3C*) and the resulting mutants were tested in STF cells co-transfected with Gpr124 and Wnt7a. Surprisingly, none of the mutants showed a significant decrement in signaling (*Figure 3D*, left panel; *Figure 3—source data 1*). One possible explanation for this result is that Reck activity depends on the cooperative actions of more amino acids than were changed in any one of the initial set of mutants. To test this idea and identify Reck mutants with subtler defects, we repeated the STF assay with two hypomorphic Wnt7a mutants (#16 and #17) that we guessed might sensitize the assay. As seen in the right two panels of *Figure 3D*, CC4 mutants #8 and #10 completely eliminated signaling specifically in the presence of Wnt7a mutants #16 and #17 (*Figure 3—source data 1*). We then combined CC4 mutants #8 and #10 to create CC4 mutant #15 (*Figure 3E*) and observed that it was completely defective in stimulating WT Wnt7a signaling in STF cells (*Figure 3G*; *Figure 3—source data 1*). Importantly, CC4 mutant #15, as well as its derivatives (#16–23) accumulate on the cell surface at levels comparable to that of WT Reck (*Figure 3F*).

Additional alanine substitution mutations among various subsets of the five residues mutated in CC4 mutant #15 identified two highly conserved residues, P256 from mutant #8 and W261 from mutant #10, that are jointly essential for signaling: simultaneous alanine substitution at these two positions (mutant #21) eliminates Reck-dependent stimulation of Wnt7a signaling in STF cells (*Figure 3E–H*; *Figure 3—source data 1*). Moreover, when P256A,W261A was introduced into Reck (CC1-5)-AP, the resulting fusion protein was unable to bind to HEK293T cells transfected with Wnt7a, Reck, and Fz5 (*Figure 3I*; summarized in 3J). Taken together, these data provide evidence for a localized region within Reck CC4 that does not affect protein stability or trafficking and is required for Wnt7a-Fz-Gpr124-Reck complex formation and signaling.

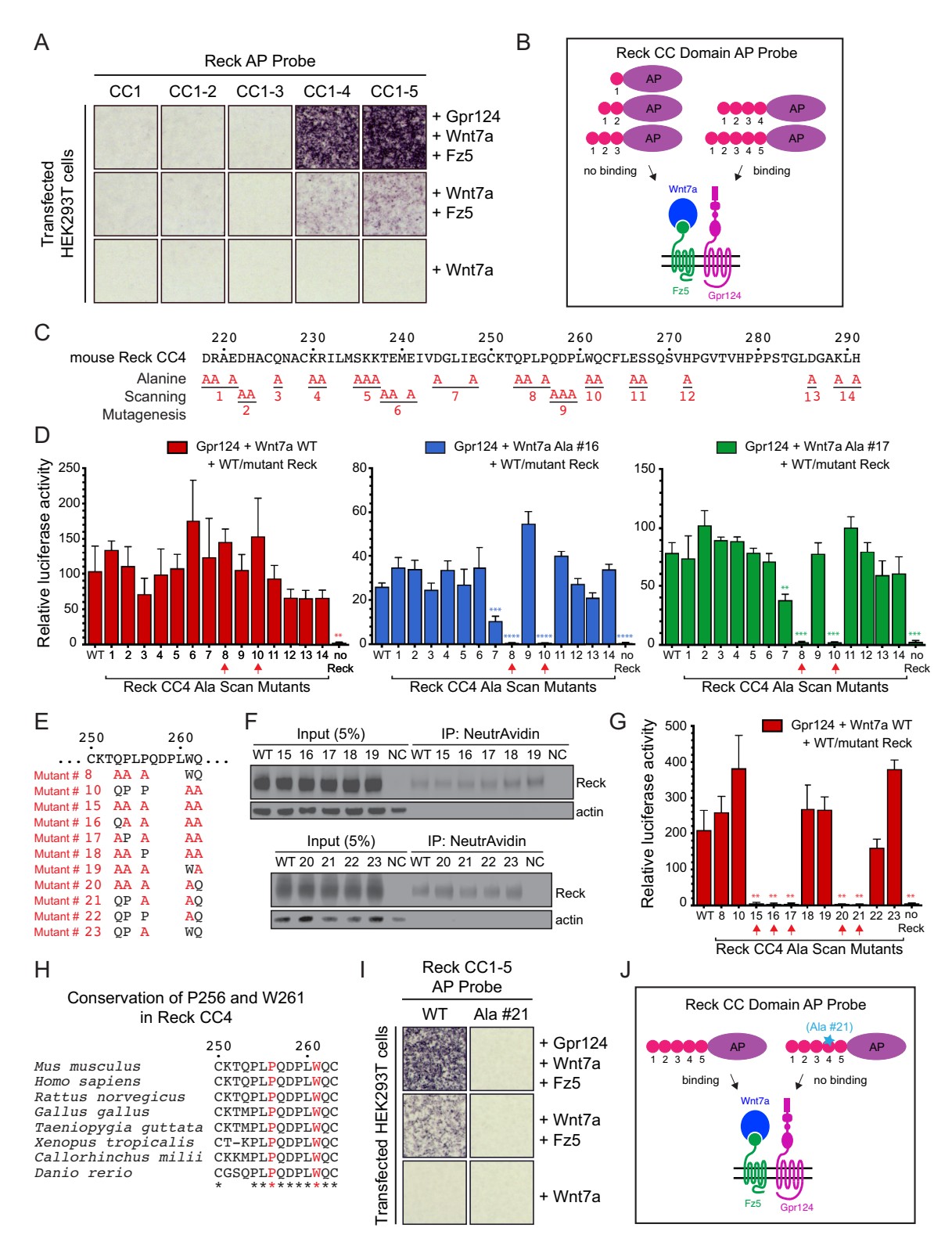

**Figure 3.** Reck CC4 is necessary for multi-protein complex formation and signaling with Gpr124, Wnt7a, and Fz. (**A**) Reck(CC1)-, (CC1-2)-, (CC1-3)-, (CC1-4)-, or (CC1-5)-AP binding to live HEK293T cells transfected as indicated at right. (**B**) Schematic of the AP binding assay in (**A**). (**C**) Amino acid sequence of mouse Reck CC4, with alanine scanning mutants indicated. (**D**) Beta-catenin signaling assay using STF cells transfected with Gpr124 and WT Reck or the indicated Reck CC4 mutant, in combination with WT Wnt7a (left), Wnt7a Ala #16 (middle), or Wnt7a Ala #17 (right). Red arrows, Reck

*Figure 3 continued on next page*

*Figure 3 continued*

CC4 Ala #8 and Ala #10 transfections. Statistical comparisons in (D) and (G) are to WT Reck. (E) Sequence of mouse Reck CC4 in the region of Ala#8 and Ala#10, with additional alanine substitution mutants indicated in red. (F) HEK293T cells were transfected with WT Reck or the indicated Reck mutants. Post-nuclear supernatants (input) and surface biotinylated proteins (captured with NeutrAvidin agarose) were immunoblotted for Reck and actin. (G) Beta-catenin signaling assay using STF cells transfected with Gpr124, Wnt7a, and WT Reck or the indicated Reck CC4 mutant. Red arrows, Reck CC4 mutants that eliminate signaling. (H) Alignment and conservation of the Reck CC4 region shown in (E) across vertebrates, generated by Clustal Omega. (*) denotes fully conserved residues. P256 and W261 are highlighted in red. (I) Reck(CC1-5)-AP and Reck(CC1-5 Ala #21)-AP binding to live HEK293T cells transfected as indicated at right. (J) Schematic of the AP binding assay in (I).
DOI: https://doi.org/10.7554/eLife.47300.009

The following source data and figure supplement are available for figure 3:

**Source data 1.** Relative luciferase activity for STF experiments in *Figure 3* panels D and G and *Figure 3—figure supplement 1* panel B.
DOI: https://doi.org/10.7554/eLife.47300.011
**Figure supplement 1.** Tests of Reck ΔCC2-3 for Wnt7a/Fz/Gpr124 signaling and complex formation.
DOI: https://doi.org/10.7554/eLife.47300.010

## Severe defects in CNS angiogenesis in embryos with a Reck CC4 mutations

To assess the effect of the P256A,W261A mutations in Reck CC4 on Wnt7a/7b signaling in vivo, these substitutions were introduced together into the mouse germline using Cas9-directed cleavage and homology-dependent repair (*Figure 4A*). The resulting $Reck^{P256A,W261A/+}$ heterozygotes are healthy and fertile, but $Reck^{P256A,W261A/P256A,W261A}$ homozygotes die by embryonic day (E)11.5. Specifically, from $Reck^{P256A,W261A/+}$ intercrosses we observed 0/42, 0/29, and 0/7 live $Reck^{P256A,W261A/P256A,W261A}$ embryos at E11.5, E12.5, and E13.5, respectively, and seven partially resorbed $Reck^{P256A,W261A/P256A,W261A}$ embryos at E11.5 and E12.5. This timing of embryonic lethality resembles that of $Reck^{\Delta ex1/\Delta ex1}$ ($\Delta ex1$ is a presumptive null allele) and both are consistent with a requirement for Reck in Wnt7b signaling and with the previously reported timing of lethality in $Wnt7b^{-/-}$ embryos, which die of placental insufficiency (*Parr et al., 2001*; *Chandana et al., 2010*).

As mid-gestational lethality precludes an analysis of CNS angiogenesis, we crossed the $Reck^{P256A,W261A}$ allele to a hypomorphic exon 2 deletion allele ($Reck^{\Delta ex2}$). $Reck^{\Delta ex2/\Delta ex2}$ mice survive to P0, albeit with severe defects in CNS angiogenesis (*Cho et al., 2017*). Conveniently, the Reck protein produced from the exon 2 deletion allele is present at very low levels in embryo extracts (*Cho et al., 2017*), thereby facilitating an assessment of the size and abundance of the $Reck^{P256A,W261A}$ protein in $Reck^{P256A,W261A/\Delta ex2}$ embryos. [*Reck* exon 2 is 59 nucleotides in length – not a multiple of three nucleotides – and codes for the first ~20 amino acids of the mature Reck protein; thus, the production of a nearly full-length protein from this allele would likely require an aberrant splicing event.] As seen in the immunoblot of E11.5 embryo extracts (*Figure 4B*), the $Reck^{P256A,W261A}$ protein exhibits a mobility and abundance that closely match those of the WT Reck protein, consistent with the properties of the $Reck^{P256A,W261A}$ protein observed in transfected cells (*Figure 3F*) and implying that any phenotype referable to the $Reck^{P256A,W261A}$ allele derives from a functional defect in CC4 rather than a defect in protein folding or stability.

In crosses between $Reck^{P256A,W261A/+}$ and $Reck^{\Delta ex2/+}$ parents, 7/69 E13.5 embryos were of the $Reck^{P256A,W261A/\Delta ex2}$ genotype, an under-representation relative to the expected 25% (p=0.04; Fisher's exact test). From the same cross, 11/50 E10.5 and E11.5 embryos had the $Reck^{P256A,W261A/\Delta ex2}$ genotype, close to the expected 25%. At E13.5, $Reck^{P256A,W261A/\Delta ex2}$ embryos exhibit intracranial bleeding and hypoplasia of the anterior limbs (*Figure 4C*). Coronal sections through E13.5 WT and $Reck^{P256A,W261A/\Delta ex2}$ brains show severe defects in angiogenesis in the cerebral cortex, striatum, and ganglionic eminences (compare panels 'a' in *Figure 4D and E*), a milder angiogenesis defect in the anterior thalamus (compare panels 'b' in *Figure 4D and E*), and normal angiogenesis in the hindbrain (compare panels 'c' in *Figure 4D and E*). In brain regions that are poorly vascularized, the neural tissue is severely hypoplastic, leading to enlargement of the ventricles. Hypovascularization is also associated with bleeding (marked by PECAM-positive red blood cells) and increased production of the glucose transporter GLUT1 in neural tissue (Figure E, panels a and b). This vascular phenotype closely matches the vascular phenotype associated with loss of function mutations in *Gpr124* (*Kuhnert et al., 2010*; *Anderson et al., 2011*; *Cullen et al., 2011*; *Zhou and Nathans, 2014*). We note that normal or nearly normal angiogenesis in the $Reck^{P256A,W261A/\Delta ex2}$ hindbrain, as well as the

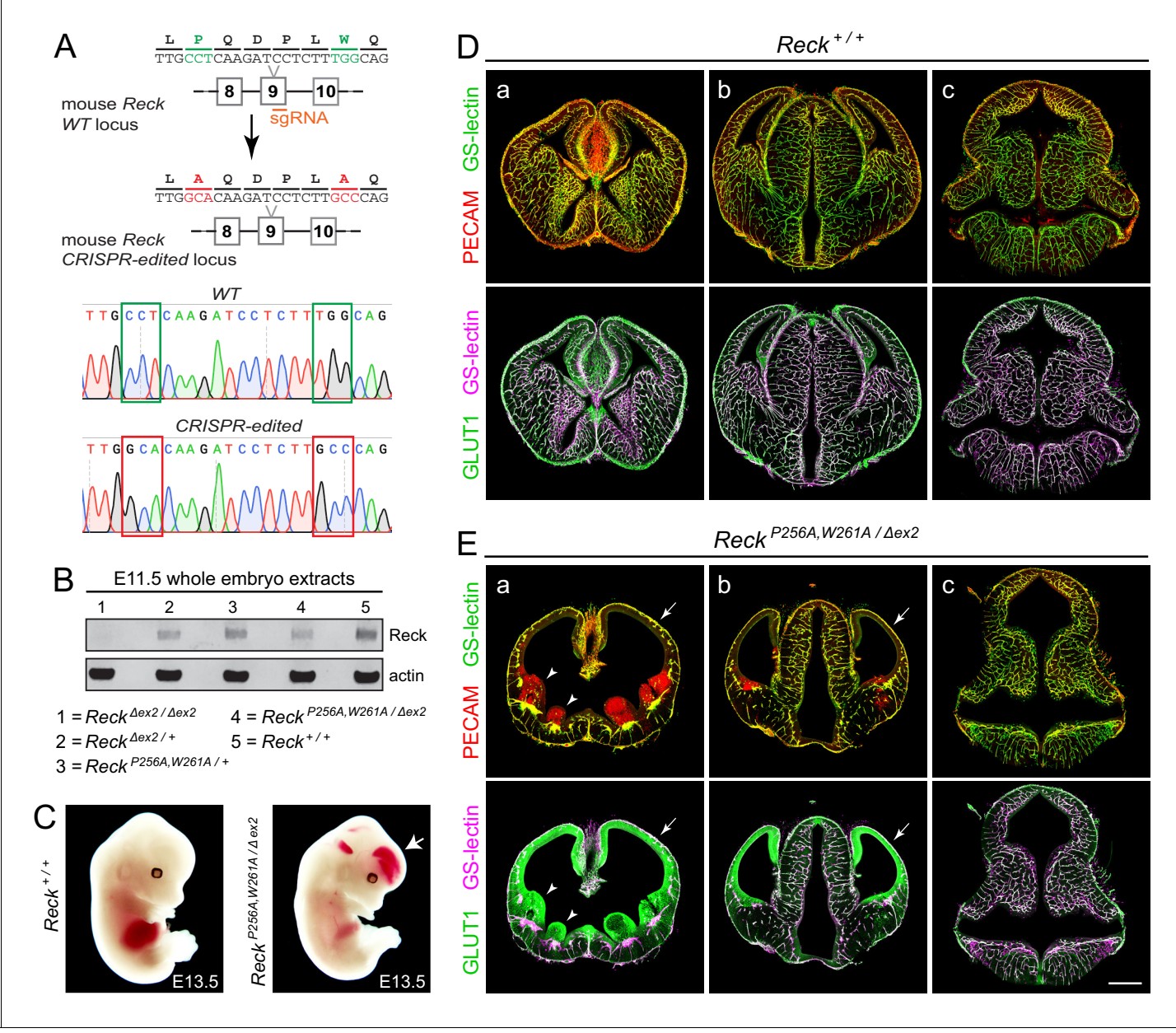

**Figure 4.** Embryos with *Reck*^P256A/W261A^ have severe defects in CNS angiogenesis that match the defects in *Gpr124* null embryos. (**A**) CRISPR/Cas9 strategy for introducing P256A and W261A into *Reck* exon 9 in the mouse germline and sequencing chromatograms from cloned genomic PCR products from WT (top) vs. *Reck*^P256A,W261A^ alleles (bottom). (**B**) Immunoblot of proteins from E11.5 embryos of the indicated genotypes, probed with anti-Reck and anti-actin antibodies. (**C**) Gross appearance of WT vs. *Reck*^P256A,W261A/Δex2^ E13.5 embryos. The arrow points to intracranial bleeding in the *Reck*^P256A,W261A/Δex2^ embryo. (**D,E**) Coronal sections of WT vs. *Reck*^P256A,W261A/Δex2^ E13.5 embryos immunostained for PECAM and GLUT1, and stained for GS-lectin. Sections are at the levels of (**a**) the ganglionic eminences, (**b**) the thalamus (center) flanked by the cerebral cortices, and (**c**) the hindbrain. In (**E**), arrows point to the avascular and hypoplastic cerebral cortex, and arrowheads point to the avascular and hypoplastic ganglionic eminences. Scale bar, 500 µm.

DOI: https://doi.org/10.7554/eLife.47300.012

nearly normal size of *Reck*^P256A,W261A/Δex2^ embryos, implies that the forebrain angiogenesis defect is not caused by general morbidity but instead reflects the local action of Reck in the developing forebrain.

In summary, the $Reck^{P256A,W261A/\Delta ex2}$ phenotype implies that (i) P256 and W261 in CC4 are essential for Reck-dependent enhancement of Wnt7a/7b signaling in vivo, and (ii) this enhancement plays an important role in CNS angiogenesis.

## Discussion

Through a combination of protein engineering, cell culture signaling, and protein binding experiments, the present work shows that the Fz CRD plays an important role in Wnt7a/Gpr124/Reck specificity and it defines two regions in Wnt7a and one region in Reck CC4 that are critical for Wnt7a/Fz/Gpr124/Reck signaling. Based on the results of these cell culture experiments and using CRISPR/Cas9 engineering in mice, we have constructed a *Reck* allele with two alanine substitutions in CC4 that produces a severe CNS vascular phenotype, providing the strongest evidence to date that Reck action in CNS angiogenesis is due to stimulation of Wnt7a/7b signaling and is independent of Reck's other function as a matrix metalloproteinase inhibitor (*Oh et al., 2001*; *Chandana et al., 2010*; *de Almeida et al., 2015*). The anatomic localization of the CNS angiogenesis defect is consistent with earlier studies showing redundancy between the Wnt7a/7b and Norrin signaling systems in the embryonic hindbrain (*Zhou and Nathans, 2014*; *Cho et al., 2017*).

As noted in the Introduction, the molecular basis of Wnt-Fz specificity is, at present, largely enigmatic. In the context of Wnt7a and Wnt7b signaling, the present work provides intriguing insights into specificity determinants in Fz receptors. In particular, the experiments in *Figure 1* show that while Wnt7a can bind to the CRDs of Fz4, Fz5, Fz6, and Fz8, only the CRDs of Fz4, Fz5, and Fz8 – but not of Fz6 – are permissive for Wnt7a/Reck/Gpr124 complex formation. Additionally, while full-length Fz4, Fz5, and Fz8 mediate beta-catenin signaling and CRD-Myc-GPI versions of Fz4, Fz5, and Fz8 inhibit beta-catenin signaling, full-length Fz6 and Fz6CRD-Myc-GPI have little or no effect on beta-catenin signaling. Finally, replacing the CRD of Fz6 with the CRD of Fz4 or Fz8 in the context of full-length Fz6 restores beta-catenin signaling to the levels obtained with full-length Fz4 or Fz8, respectively. The simplest explanation for all of these observations is that Fz specificity in this system is mediated by the CRD, presumably via direct contact with Reck and/or Gpr124.

It is interesting that the Fz4CRD-Fz6 chimera cannot elicit a beta-catenin response to Norrin despite robust binding, suggesting that regions of Fz4 beyond the CRD are required for signal transmission in the Norrin-Fz4 system. Our observations with Norrin and Frizzled (*Figure 1F and G*) are consistent with those from a recent study from *Bang et al. (2018)*, which has identified the linker region between the Fz4 CRD and transmembrane domain, as well as sites in intracellular loop 3, as important for enhanced Norrin binding and signaling.

In the context of CNS vascular development, the expression of Fz4, Fz5, and Fz8 in ECs, together with their competence for Wnt7a/Gpr124/Reck signaling, implies that they are the mediators of Wnt7a and Wnt7b signaling in vivo. The role of Fz6 in vascular development remains unclear, despite its robust expression in ECs, as $Fz6^{-/-}$ mice show no apparent vascular phenotypes (*Wang et al., 2006*).

With respect to specificity determinants in Wnt7a, the present work complements recent studies from *Eubelen et al. (2018)* that independently identified the protruding region of Wnt7a (encompassing alanine substitutions #15 and #18) as critical for Wnt7a/Fz/Gpr124/Reck signaling based on substitution mutations that primarily targeted large hydrophobic amino acids. *Eubelen et al. (2018)* have also presented evidence that Reck(CC1-5) can bind with low micromolar affinity to a synthetic peptide corresponding to this region of Wnt7a or Wnt7b, and that CC4 and CC5 are required for that interaction. Our observation that Reck(CC1-5)-AP carrying the P256A,W261A substitutions is unable to participate in Wnt7a/Fz/Gpr124 complex formation is consistent with the model of Eubelen et al., but it is also consistent with alternate models for CC4 function.

The present work additionally complements the binding studies performed by *Vallon et al. (2018)* that demonstrate a direct interaction between Reck and Wnt7a. Intriguingly, Vallon et al. showed that a soluble Reck-Fc fusion protein stimulates binding of secreted Wnt7a to a soluble Fz8 CRD protein, suggesting a model in which Reck, in conjunction with Gpr124, presents Wnt7a and Wnt7b to Fz receptors at the plasma membrane. While a full understanding of Wnt7a/Wnt7b/Fz/Gpr124/Reck/Lrp ligand recognition and signaling will require high-resolution structural information, the biochemical and functional studies to date, including the present work, substantially

constrain current models by defining essential protein-protein interactions and the domains that mediate them.

# Materials and methods

## Key resources table

| Reagent type (species) or resource | Designation | Source or reference | Identifiers | Additional information |
|---|---|---|---|---|
| Genetic reagent (*M. musculus*) | *Reck$^{\Delta ex2}$* | PMID: 20691046 | RRID:MGI:4830344 | |
| Genetic reagent (*M. musculus*) | *Reck$^{P256A,W261A}$* | this paper | | Please find details under Materials and methods (Gene Targeting) |
| Cell line (*H. sapiens*) | HEK/293T | ATCC | Cat. #: CRL-3216; RRID:CVCL_0063 | |
| Cell line (*H. sapiens*) | Super TOP Flash (STF) luciferase reporter cell line | PMID: 15035989 | | |
| Antibody | Rabbit polyclonal anti-Glut1 | Thermo Fisher Scientific | Cat. #: RB-9052-P1; RRID: AB_177895 | 1:400 dilution |
| Antibody | Rat monoclonal anti-PECAM/CD31 | BD Biosciences | Cat. #: 553370; RRID: AB_394816 | 1:400 dilution |
| Antibody | Isolectin GS-IB4 (GS Lectin), Alexa 488 conjugate | Thermo Fisher Scientific | Cat. #: I21411, RRID: AB_2314662 | 1:400 dilution |
| Antibody | Rabbit polyclonal anti-6xMyc | PMID: 28803732 | | 1:10,000 dilution |
| Antibody | Rat monoclonal anti-alpha tubulin | Thermo Fisher Scientific | Cat# MA1-80017; RRID: AB_2210201 | 1:10,000 dilution |
| Antibody | Mouse monoclonal anti-actin | Millipore Sigma | Cat. #: MAB1501; RRID: AB_2223041 | 1:10,000 dilution |
| Antibody | Rabbit monoclonal anti-Reck | Cell Signaling | Cat. #: 3433S; RRID: AB_2238311 | 1:2000 dilution |
| Antibody | Alkaline phosphatase horse anti-mouse IgG antibody | Vector Laboratories | Cat. #: AP-2000; RRID:AB_2336173 | 1:10,000 dilution |
| Antibody | Goat polyclonal anti-rabbit IgG (H + L) cross-adsorbed secondary antibody, Alexa 488, 594, and 647 conjugates | Thermo Fisher Scientific | Cat. #s: A-11008, RRID: AB_143165; A-11012, RRID: AB_2534079; A-21244, RRID: AB_2535812 | 1:400 dilution |
| Antibody | Goat polyclonal anti-rat IgG (H + L) cross-adsorbed secondary antibody, Alexa 488, 594, and 647 conjugates | Thermo Fisher Scientific | Cat. #s: A-11006, RRID: AB_2534074; A-11007, RRID: AB_2534075; A-21247, RRID: AB_141778 | 1:400 dilution |
| Antibody | IRDye 800CW goat anti-mouse IgG (H + L) secondary antibody | LI-COR | Cat. #: 925–32210; RRID:AB_2687825 | 1:10,000 dilution |

*Continued on next page*

*Continued*

| Reagent type (species) or resource | Designation | Source or reference | Identifiers | Additional information |
|---|---|---|---|---|
| Antibody | IRDye 680RD goat anti-rabbit IgG (H + L) secondary antibody | LI-COR | Cat. #: 925–68071; RRID:AB_2721181 | 1:10,000 dilution |
| Antibody | IRDye 680RD goat anti-rat IgG (H + L) secondary antibody | LI-COR | Cat. #: 926–68076; RRID:AB_10956590 | 1:10,000 dilution |
| Oligonucleotides | *Reck*^P256A,W261A guide RNA: caagatcctctttggcagtg | this paper | | Please find details under Materials and methods (Gene Targeting) |
| Oligonucleotides | *Reck*^P256A,W261A SSODN HDR template: gttgatggtctcattgagggttgta agacccagcccttggcacaagatc ctcttgcccagtgttttctc gaaagctcacagtc ggttcaccctgga | this paper | | Please find details under Materials and methods (Gene Targeting) |
| Recombinant DNA reagents | Mouse Frizzled CRD-GPI cDNA | PMID: 17158104 | | |
| Recombinant DNA reagents | Mouse Norrin, Wnts, and Frizzleds cDNA | PMID: 23095888 | | |
| Recombinant DNA reagents | Mouse Tspan12 cDNA | PMID: 30478038 | | |
| Recombinant DNA reagents | Mouse Reck cDNA | PMID: 28803732 | | |
| Recombinant DNA reagents | Mouse Gpr124 cDNA | PMID: 28803732 | | |
| Recombinant DNA reagents | Frizzled chimera cDNA | this paper | | Please find details under Materials and methods (Plasmids) |
| Recombinant DNA reagents | Wnt7a chimera and mutant cDNA | this paper | | Please find details under Materials and methods (Plasmids) |
| Recombinant DNA reagents | Reck mutant cDNA | this paper | | Please find details under Materials and methods (Plasmids) |
| Recombinant DNA reagents | Reck AP fusion cDNA | this paper | | Please find details under Materials and methods (Plasmids) |
| Commercial assay or kit | Dual-Luciferase Reporter Assay System | Promega | Cat. #: E1910 | |
| Chemical compound, drug | BluePhos phosphatase substrate solution (5-bromo-4-chloro-3-indolyl phosphate/tetrazolium) | Kirkegaard and Perry Laboratories | Cat. #: 50-88-00 | |
| Chemical compound, drug | EZ-Link Sulfo-NHS-LC-Biotin | Thermo Fisher Scientific | Cat. #: 21335 | |
| Chemical compound, drug | Nitro blue tetrazolium/ 5-bromo-4-chloro-3-indolyl phosphate (NBT/BCIP) substrate | Roche | Cat. #: 11383213001 | |

*Continued on next page*

*Continued*

| Reagent type (species) or resource | Designation | Source or reference | Identifiers | Additional information |
|---|---|---|---|---|
| Software, algorithm | ImageJ | https://imagej.nih.gov/ij | | |
| Software, algorithm | Adobe Photoshop CS6 | https://adobe.com/photoshop | | |
| Software, algorithm | Adobe Illustrator CS6 | https://adobe.com/illustrator | | |
| Software, algorithm | GraphPad Prism 7 | http://www.graphpad.com | | |
| Other | FuGENE HD Transfection Reagent | Promega | Cat. #: E2311 | |
| Other | Pierce NeutrAvidin agarose resin | Thermo Fisher Scientific | Cat. #: 29200 | |
| Other | Fluoromount G | EM Sciences | Cat. #: 17984–25 | |
| Other | Protease Inhibitor | Roche | Cat. #: 11836170001 | |

## Gene targeting

The $Reck^{P256A,W261A}$ mouse was generated using CRISPR/Cas9 gene editing. An Alt-R CRISPR-Cas9 crRNA (caagatcctctttggcagtg) targeting exon 9 of *Reck* was selected and synthesized by Integrated DNA Technologies (IDT). The ssODN HDR template (gttgatggtctcattgagggttgtaagacccagcccttggcacaagatcctcttgcccagtgttttctcgaaagctcacagtcggttcaccctgga) was synthesized by IDT. The crRNA, tracrRNA, ssODN HDR template, and Cas9 protein were injected into C57BL/6 x SJL F2 embryos to generate correctly targeted founders.

The $Reck^{P256A,W261A}$ allele was genotyped by PCR with the following primers: gcacaagatcctcttgcc (Forward) and gcccgtaactccaactccag (Reverse) with an expected product of 474 base pairs. The corresponding wild type allele was genotyped by PCR with the following primers: cctcaagatcctctttggc (Forward) and gcccgtaactccaactccag (Reverse) with an expected product of 474 base pairs. PCR conditions were as follows: 94°C, 4 min; 94°C, 30 sec / 60°C, 30 sec / 72°C, 30 s for 35 cycles; 72°C, 10 min.

## Mice

The following mouse alleles were used: $Reck^{\Delta ex2}$ (*Chandana et al., 2010*) and $Reck^{P256A,W261A}$ (this paper). All mouse experiments were performed according to the approved Institutional Animal Care and Use Committee (IACUC) protocol MO16M369 of the Johns Hopkins Medical Institutions.

## Antibodies and other reagents

The following antibodies were used for tissue immunohistochemistry: rat anti-mouse PECAM/CD31 (BD Biosciences 553370); rabbit anti-GLUT1 (Thermo Fisher Scientific RB-9052-P1). Alexa Fluor-labeled secondary antibodies and GS Lectin (Isolectin GS-IB4) were from Thermo Fisher Scientific.

The following antibodies were used for immunoblot and immuno-staining analysis: mouse anti-1D4 (*MacKenzie et al., 1984*); rabbit anti-6xMyc (JH6204); rabbit anti-Reck (Cell Signaling 3433); rat anti-alpha tubulin (Invitrogen MA1-80017); and AP-conjugated horse anti-mouse IgG antibody (Vector Laboratories AP-2000). Fluorescent secondary antibodies for immunoblotting were from Li-Cor.

## Tissue processing and immunohistochemistry

Tissues were prepared and processed for immunostaining analysis as described by *Wang et al. (2012)* and *Zhou et al. (2014)*. Briefly, embryos were harvested and immersion fixed overnight at 4°C in 1% PFA, followed by 100% MeOH dehydration overnight at 4°C. All tissues were re-hydrated the following day in 1x PBS at 4°C for at least 3 hr before embedding in 3% agarose. Tissue sections of 150–180 μm thickness were cut using a vibratome (Leica).

Sections were incubated overnight with primary antibodies (1:400) diluted in 1x PBSTC (1x PBS + 0.5% Triton X-100 +0.1 mM $CaCl_2$) + 10% normal goat serum (NGS). Sections were washed at

least 3 times with 1x PBSTC over the course of 6 hr, and subsequently incubated overnight with secondary antibodies (1:400) diluted in 1x PBSTC + 10% NGS. If a primary antibody raised in rat was used, secondary antibodies were additionally incubated with 1% normal mouse serum (NMS) as a blocking agent. The next day, sections were washed at least 3 times with 1x PBSTC over the course of 6 hr, and flat-mounted using Fluoromount G (EM Sciences 17984–25). Sections were imaged using a Zeiss LSM700 confocal microscope, and processed with ImageJ, Adobe Photoshop, and Adobe Illustrator software. Incubation and washing steps were performed at 4°C.

## Plasmids

The Fz chimeras were generated by PCR amplification of the Fz4 CRD (aa1-169) and Fz8 CRD (aa1-160) and cloned into full length Fz6 to replace the Fz6 CRD (aa1-135). The Wnt chimeras were generated by PCR amplification of the N-terminal domains of Wnt7a (aa1-268), Wnt3 (aa1-274), and Wnt3a (aa1-271) and the C-terminal domains of Wnt7a (aa269-349), Wnt3 (aa275-355), and Wnt3a (aa272-352). Alanine mutagenesis of Wnt7a and Reck was performed by tandem PCR. Inserts for AP constructs were PCR amplified from a Reck cDNA clone. Reck ΔCC2-3 constructs were generated by replacing aa97-203 with a short (GG) or long (GSGGSGGSG) spacer. The expression plasmid for AP-3xMyc-Norrin is described in *Xu et al. (2004)*. Expression plasmids for the mouse Wnts, Frizzleds, and FzCRD-Myc-GPIs are described in *Smallwood et al. (2007)* and *Yu et al. (2012)*.

## Cell lines

HEK/293T cells (ATCC CRL-3216) and Super TOP Flash (STF) cells (*Xu et al., 2004*) were used in this study, and there was no evidence of mycoplasma contamination. We confirmed cell line identity by RNA sequencing. Cells were grown in DMEM/F-12 supplemented with 10% fetal bovine serum (FBS) and passaged at a dilution of 1:5 for no more than a maximum of 20 passages. HEK/293T cells were seeded into 6-well or 12-well plates at a confluency of 70–80% prior to transfection. STF cells were seeded into 96-well plates at a confluency of 30–40% prior to transfection. Experimental details are further elaborated below in 'Luciferase assays', 'Alkaline Phosphatase binding assays', and 'Cell surface biotinylation assay and immunoblot analysis'.

## Luciferase assays

Dual luciferase assays were performed as described by *Xu et al. (2004)*. Briefly, STF cells were plated on 96-well plates at a confluency of 30–40%. The following day, fresh DMEM/F-12 (Thermo Fisher Scientific 12500) supplemented with 10% fetal bovine serum (FBS) replaced the medium in each of the wells. Three hours later, cells were transfected in triplicate with expression plasmids (180–240 ng of DNA per three wells) using FuGENE HD Transfection Reagent (Promega E2311). The DNA master mix included: 1.5 ng of the internal control Renilla luciferase plasmid (pRL-TK), and 60 ng each of the pRK5 expression plasmids for Gpr124, Reck, Wnt7a, Fz, and control vector. 48 hr post-transfection, cells were harvested in 1x Passive Lysis Buffer (Promega E194A) for 20 min at room temperature. Lysates were used to measure Firefly and Renilla luciferase activity using the Dual-Luciferase Reporter Assay System (Promega E1910) and a Turner BioSystems Luminometer (TD-20/20). Relative luciferase activity was calculated by normalizing Firefly/Renilla values. GraphPad Prism 7 software was used to generate plots and perform statistical analysis. The mean ± standard deviations are shown.

## Alkaline Phosphatase binding assays

AP binding assays were performed as described by *Cho et al. (2017)*. Briefly, conditioned DMEM/F-12 media containing AP fusion proteins were collected from HEK293T cells that were transfected with pRK5 expression plasmids using FuGENE HD Transfection Reagent. All conditioned media were collected 72 hr post-transfection and spun-down to remove detached cells. To quantify the yield of AP fusion proteins, an aliquot of the conditioned medium was incubated with BluePhos phosphatase substrate solution (5-bromo-4-chloro-3-indolyl phosphate/tetrazolium; Kirkegaard and Perry Laboratories 50-88-00).

For cell-based AP binding assays to assess multi-protein complexes, HEK293T cells were plated on 0.2% gelatin coated wells. 48 hr post-transfection, cells were incubated with serum-containing conditioned medium at 4°C for 2 hr. Cells were washed 3 times with cold serum-free DMEM/F-12

medium prior to fixing with cold 4% paraformaldehyde (PFA) in PBS. Fixed cells were placed in a 70°C water bath for 1 hr to heat denature endogenous AP. Bound AP was visualized by incubation with nitro blue tetrazolium/5-bromo-4-chloro-3-indolyl phosphate (NBT/BCIP) substrate (Roche 11383213001) at room temperature.

For cell-surface AP immuno-staining assays, HEK293T cells were plated on 0.2% gelatin coated wells. 48 hr post-transfection, cells were incubated with diluted primary antibodies (1:1,000) in serum-containing media at 4°C for 1 hr, and then washed 5–6 times with cold PBS. Cells were subsequently fixed and heat denatured as described above. Cells were incubated with diluted AP-conjugated horse anti-mouse IgG antibody (1:1000) in 1x PBS at 4°C for 1 hr. Cells were washed 3 times with 1x PBS prior to AP visualization as described above.

### Cell surface biotinylation assay and immunoblot analysis

Cell surface biotinylation was performed as described by Pavel et al. (2014). In brief, HEK293T cells were plated on 0.2% gelatin coated wells and transfected with pRK5 expression plasmids with FuGENE HD Transfection Reagent. The medium was removed 48 hr post-transfection, and cells were washed 3 times with 1x PBS. Cells were incubated in 1x PBS containing Sulfo-NHS-Biotin (250 µg/ml) at 4°C for 30 min. Excess biotin was quenched by adding Tris-HCl pH 7.4 to a final concentration of 50 mM at 4°C for 5 min. After removing the Tris buffer, cells were detached, washed 3 times in 1x Tris buffered saline (TBS), and lysed in 1x RIPA buffer (50 mM Tris-HCl pH 7.4, 150 mM NaCl, 1% Triton X-100, and 0.5% deoxycholate) containing protease inhibitor (Roche 11836170001). Cell lysates were incubated at 4°C for 30 min and subsequently centrifuged at 10,000xg at 4°C for 20 min to remove cellular debris. Cell-surface proteins were captured by incubating cleared lysates with NeutrAvidin Agarose Resin (Thermo Fisher Scientific 29200) overnight at 4°C. Resin was washed 5 to 6 times with 1x RIPA buffer, and captured proteins, along with input controls, were resolved by SDS-PAGE and transferred to PVDF membranes (EMD Millipore IPFL00010) for immunoblotting. Immunoblots were incubated with primary antibodies (1:10,000 mouse anti-1D4, mouse anti-actin, and rat anti-alpha tubulin; 1:2,000 rabbit anti-Reck) diluted in Odyssey Blocking Buffer (LiCor 927–40000) overnight at 4°C. Membranes were washed with 1x PBS-T (1x PBS + 0.1% Tween 20), and incubated with LiCor secondary antibodies (1:10,000) diluted in Odyssey Blocking Buffer for 1 hr at room temperature. Membranes were washed at least 3 times with 1x PBS-T and developed using the Odyssey Fc Imaging System (LiCor).

For whole tissue lysates, E11.5 embryos were harvested and homogenized using a plastic pestle in 1x RIPA buffer supplemented with protease inhibitors. Tissue lysates were cleared by centrifugation, and the supernatants processed for immunoblot analysis, as described above.

### Modeling Wnt7a structure

The mouse Wnt7a (mWnt7a) structure was modeled by using the Xenopus Wnt8 (XWnt8) crystal structure in complex with mouse Fz8 CRD (Protein Data Bank code 4F0A; Janda et al., 2012) using The PyMOL Molecular Graphics System, Version 2.2.3 Schrödinger, LLC. The amino acid sequence of mWnt7a (amino acids (aa)54–349) aligns with XWnt8 (aa32-338) with four aa insertions in XWnt8 (Y37, L38, T39, and Y40) that are not present in mWnt7a.

### Quantification and statistical analysis

GraphPad Prism 7 software was used to generate plots and to perform statistical analysis. The mean ± standard deviations are shown. Statistical significance was determined by the unpaired t-test, and is represented by * ($p < 0.05$), ** ($p < 0.01$), *** ($p < 0.001$), and **** ($p < 0.0001$).

The analysis of alignment and conservation of Reck CC4 across vertebrates was generated using Clustal Omega (McWilliam et al., 2013).

## Acknowledgements

The authors thank Ann Lawler and Chip Hawkins from the Johns Hopkins Transgenic Core lab for CRISPR/Cas9 injection, and Tao-Hsin Chang, Amir Rattner, and Mark Sabbagh for advice and helpful comments on the manuscript. Supported by the Howard Hughes Medical Institute, the National Eye Institute (NIH) (R01EY018637), and the Arnold and Mabel Beckman Foundation.

## Additional information

### Competing interests

Jeremy Nathans: Reviewing editor, *eLife*. The other authors declare that no competing interests exist.

### Funding

| Funder | Grant reference number | Author |
|---|---|---|
| Howard Hughes Medical Institute | | Jeremy Nathans |
| National Eye Institute | R01EY018637 | Jeremy Nathans |
| Arnold and Mabel Beckman Foundation | | Jeremy Nathans |

The funders had no role in study design, data collection and interpretation, or the decision to submit the work for publication.

### Author contributions

Chris Cho, Conceptualization, Data curation, Formal analysis, Validation, Investigation, Methodology, Writing—original draft, Writing—review and editing; Yanshu Wang, Data curation, Formal analysis, Validation, Methodology; Philip M Smallwood, Data curation, Formal analysis, Validation; John Williams, Data curation, Validation; Jeremy Nathans, Conceptualization, Resources, Data curation, Software, Formal analysis, Supervision, Funding acquisition, Investigation, Writing—original draft, Project administration, Writing—review and editing

### Author ORCIDs

Chris Cho (iD) https://orcid.org/0000-0002-0929-6536
Jeremy Nathans (iD) https://orcid.org/0000-0001-8106-5460

### Ethics

Animal experimentation: All mice were housed and handled according to the approved Institutional Animal Care and Use Committee (IACUC) protocol MO16M369 of the Johns Hopkins Medical Institutions.

### Decision letter and Author response

Decision letter https://doi.org/10.7554/eLife.47300.015
Author response https://doi.org/10.7554/eLife.47300.016

## Additional files

### Supplementary files

• Transparent reporting form
DOI: https://doi.org/10.7554/eLife.47300.013

### Data availability

All data generated or analyzed during this study are included in the manuscript and supporting files. Source data files have been provided for Figures 1, 2, and 3.

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
