## [Decision Letter]

Thank you for submitting your article "Molecular determinants in Reck and Wnt7a for ligand-specific β-catenin signaling in neurovascular development" for consideration by *eLife*. Your article has been reviewed by three peer reviewers, including Karl Willert as the guest Reviewing Editor and Reviewer #1, and the evaluation has been overseen by Didier Stainier as the Senior Editor. The following individuals involved in review of your submission have also agreed to reveal their identity: Hsin-Yi Henry Ho (Reviewer #2) and Stefan Liebner (Reviewer #3).

The reviewers have discussed the reviews with one another and the Reviewing Editor has drafted this decision to help you prepare a revised submission.

Summary:

This manuscript addresses an important question in developmental biology, namely how is signaling specificity achieved. The mammalian genome encodes 19 Wnt signaling molecules and 10 Frizzled (Fz) receptors, however, to date relatively little is known about how specific ligand (Wnt) – receptor (Fz) pairs regulate downstream developmental processes. Here, the authors examine the role of Wnt signaling in the development of the vasculature in the CNS, a process that involves the molecular interaction of Wnt7a/b with the Reck-Gpr124-Fzd-Lrp5/6 receptor complex. By generating a series of mutations in both Reck and Wnt7a, they are able to identify small stretches of amino acids that are required for assembly of this complex and subsequent downstream signaling. To ascertain the in vivo significance of a particular sequence motif (involving P256A/W261A) within the CC4 domain of Reck, the authors generated a novel mouse model carrying P256A/W261A point mutations and showed that these mutations cause angiogenic defects when expressed in combination with a hypomorphic Reck (deleted exon2) allele. This manuscript complements previous reports by Eubelen et al., 2018 and Vallon et al., 2018, which have independently shown that Reck mediates specificity for Wnt7 binding and signaling to the receptor complex in endothelial cells. Importantly, this current work builds on these molecular findings by providing the best data yet that specific interactions between Wnt7 and Reck are required for CNS angiogenesis.

Essential revisions:

1) Reck-Wnt7 interaction: The data presented here argue strongly for a direct interaction between Reck and Wnt7. The prior work by Eubelen et al. indicated that a Wnt7 peptide interacted with Reck with low affinity. The P256A, W261A Reck protein is expected to disrupt Wnt7 binding. The authors should provide biochemical evidence for this loss of interaction.

2) Clarification and further characterization of the ex2 allele of Reck. The authors state that "the Reck protein produced from the exon 2 deletion allele is present at very low levels in embryo extracts", citing their prior publication (Cho et al., 2017). However, a figure provided in this prior publication (Figure 6E in the 2017 paper) shows no protein on an immunoblot, similar to what is shown here in Figure 4B. Furthermore, the original paper that describes the generation of this Reck allele (Chandana et al., 2010) provides no evidence of low levels of Reck protein expression. Since this current study relies on this hypomorphic allele to analyze the E15.5 phenotype of the newly generated P256A, W261A allele (phenotypically a null allele and embryonic lethal at E11.5 when homozygous), the authors need to provide additional characterization of this reagent. Currently, the hypomorph description is based on phenotype and the fact that animals develop to E15.5, but there is no clear evidence that any Reck protein is made, as indicated by the authors.

3) Characterization of Reck-CC4: Consistent with other studies, CC4 is required for complex formation, however, is it sufficient? The authors' earlier work (Cho et al., 2017) showed that CC1 is also essential by directly binding Gpr124, so most likely both CC1 and CC4 are required for complex formation. It would be interesting to learn if a protein containing only CC1 and CC4 is sufficient for complex formation and signaling by bridging Gpr124 and Wnt7.

4) Norrin interactions (Figure 1F,G): In the context of the provided data on norrin binding and signaling, the authors should cite and discuss the manuscript by Bang et al., 2018, which may explain the finding that Fzd4CRD-Fzd6 did not signal. Specifically, a flexible linker domain between the CRD and TM1 in Fzd4 plays an important role in Norrin signaling, and inclusion of this linker in the Fz4CRD-Fz6 construct may rescue the lack of β-catenin signaling.

---

## [Author Response]

Essential revisions:1) Reck-Wnt7 interaction: The data presented here argue strongly for a direct interaction between Reck and Wnt7. The prior work by Eubelen et al. indicated that a Wnt7 peptide interacted with Reck with low affinity. The P256A, W261A Reck protein is expected to disrupt Wnt7 binding. The authors should provide biochemical evidence for this loss of interaction.

Our data and that of Eubelen et al., 2018, are mutually reinforcing in that both demonstrate an important role for the protruding “knob” domain of Wnt7a. Furthermore, Eubelen et al. showed that a peptide corresponding to this Wnt7a or Wnt7b sequence binds to a Reck CC domain-Fc fusion with low affinity (1-10 uM). As noted by the editor and the reviewers, our identification of a small region of Reck CC4 that is critical for cell-surface ligand-receptor complex formation and signaling suggests, as the most parsimonious explanation, that this CC4 region directly contacts the Wnt7a “knob”. In experiments that preceded the initial submission, we attempted to detect such an interaction in two ways, but neither experiment produced a signal over background.

First, we produced the Wnt7a knob region displayed as an Fc fusion (i.e. a dimeric fusion) and probed it with Reck(CC1-5)-AP (also a dimer). For this analysis, we expressed both a linear version of the Wnt7a “knob” region and a version in which we engineered a pair of cysteines flanking the Wnt7a “knob” sequences so that the resulting disulfide bonded loop would create a locally constrained structure similar to the one seen in the Wnt-CRD crystal structure determined by Janda et al., 2012. Although the Fc-dimer-to-AP-dimer binding strategy facilitates the detection of relatively weak interactions and the colorimetric AP detection method is highly sensitive, the binding signals for these experiments were below the limit of detection. We have also tested Reck(CC1-5)-AP binding to cell surface complexes with the Wnt7a alanine substitution mutants (including the ones in the “knob” region) and we did not observe a correlation between reduced binding and reduced signaling in STF cells.

Second, we co-transfected full-length Wnt7a-1D4 with Reck(CC1-5)-Fc or Reck(CC1-5+CRD)-Fc [Reck has a Frizzled-like CRD motif C-terminal to the CC domains], captured the Fc fusion proteins from serum-free conditioned medium, and assessed the level of bound Wnt7a-1D4 by immunoblotting. These experiments were performed with WT vs. P256A,W261A versions of the Reck CC4 sequence. With both constructs, we observed no differences between WT vs. P256A,W261A in the levels of bound Wnt7a. For both WT and mutant, the capture is more efficient when the CRD is present. In contrast to the results of these binding experiments using soluble Wnt and Reck proteins, we see a dramatic difference between WT vs. P256A,W261A when we assay Reck(CC1-5)-AP binding to living cells displaying Fz+Gpr124+Wnt7a (shown in Figure 3I).

What is our interpretation at this point? First, we think that the Wnt7a “knob” region (as well as the Wnt7a N-terminal region, which is also important for STF signaling) may be doing something more than mediating binding to Reck(CC1-5), but we do not know what that is. Second, in characterizing mutations in Reck(CC1-5), we think that the cell surface Reck(CC1-5)-AP binding assay provides biologically relevant data because: (1) all of the receptor/ligand components are present and all are required for optimal binding, (2) the binding occurs in the living (i.e. native and unfixed) cell surface environment in which signaling occurs, and it includes additional components like proteoglycans and lipids, and (3) this assay shows that the P256A,W261A mutant is severely defective, in agreement with its complete loss of function in the STF signaling assay and in vivo.

We have not presented the “negative” binding results described above because we are not convinced that those are definitive experiments. For example, the sensitivity of our Wnt7a “knob”-Fc binding experiment may be lower than the sensitivity of the synthetic peptide binding assays in Eubelen et al. (which used isothermal titration calorimetry), so we do not want to leave the reader with the impression that we have failed to replicate the Eubelen et al. data. Our current view is that Wnt7a recognition by Reck and Gpr124 normally occurs in the context of a multi-protein complex, and, as a result, some of the critical interactions are not easily demonstrated with isolated components or with protein fragments.

2) Clarification and further characterization of the ex2 allele of Reck. The authors state that "the Reck protein produced from the exon 2 deletion allele is present at very low levels in embryo extracts", citing their prior publication (Cho et al., 2017). However, a figure provided in this prior publication (Figure 6E in the 2017 paper) shows no protein on an immunoblot, similar to what is shown here in Figure 4B. Furthermore, the original paper that describes the generation of this Reck allele (Chandana et al., 2010) provides no evidence of low levels of Reck protein expression. Since this current study relies on this hypomorphic allele to analyze the E15.5 phenotype of the newly generated P256A, W261A allele (phenotypically a null allele and embryonic lethal at E11.5 when homozygous), the authors need to provide additional characterization of this reagent. Currently, the hypomorph description is based on phenotype and the fact that animals develop to E15.5, but there is no clear evidence that any Reck protein is made, as indicated by the authors.

By way of background, the Reck floxed exon 2 mutant was created by Noda’s group approximately ten years ago and is described in Chandana et al., 2010. The germline deleted version of this allele (“Reck exon2del”) is phenotypically hypomorphic. Reck exon 2 is 59 nucleotides in length and it codes for the first 20 amino acids after the signal peptide. This exon is not a multiple of 3 nucleotides, and therefore, its deletion will lead to a frame-shift. The fact that Reck exon2del is phenotypically hypomorphic, rather than null, implies that a cryptic splicing event or frame-shifting translation is allowing a polypeptide product to be produced. The mouse mAb that we have used for western blotting is a rabbit mAb (from Cell Signaling Technology) that was raised against the C-terminal region, i.e. a region that is not encoded by exon2 and is presumably present in the Reck exon2del protein. The hypomorphism of the Reck exon2del allele and the very low level of the Reck exon2del protein allowed us to (1) study compound heterozygous exon2del and P256A,W261A embryos beyond the age when homozygous P256A,W261A embryos would have died, and (2) observe the P256A,W261A protein produced by compound heterozygous embryos on western blots.

With respect to the specific point raised above – whether the Reck exon2del protein is “absent” or “present at very low level” – the hypomorphic phenotype implies the latter. If the Reck exon2del protein were “absent”, then this allele would be a null. With standard image processing of our western blots, the Reck exon2del protein appears to be “absent” – or, stated more precisely, it is “below the limit of detection under standard conditions”. However, if we electronically increase the signal we see a faint band at a molecular weight a bit lower than that of the WT Reck protein (red arrow in the enclosed image). Similarly, Chandana et al. show a western blot of a CreER mediated deletion of the Reck exon2 floxed allele (see their Figure 5B), which they describe in the legend as showing “the absence of Reck protein band” – in fact, there is a faint band at a bit lower molecular weight. We do not want to draw a firm conclusion from very faint bands, but the data suggest that this faint band may correspond to the Reck exon2del protein. We have added a sentence to the Results to clarify this issue.

The important point for this analysis is that the very low level of the Reck exon2del protein allows us to observe the Reck P256A,W261A protein on western blots from compound heterozygous embryos.

3) Characterization of Reck-CC4: Consistent with other studies, CC4 is required for complex formation, however, is it sufficient? The authors' earlier work (Cho et al., 2017) showed that CC1 is also essential by directly binding Gpr124, so most likely both CC1 and CC4 are required for complex formation. It would be interesting to learn if a protein containing only CC1 and CC4 is sufficient for complex formation and signaling by bridging Gpr124 and Wnt7.

Thank you for that suggestion. Over the past month, we tested this idea with two Reck mutants, in which CC2 and CC3 were deleted: one with a small gly/ser linker and one with a larger gly/ser linker connecting CC1 and CC4. The interesting result is that the two mutants support a modest level of signaling (4-6 fold lower than WT Reck), and when expressed as AP fusion proteins, they confer nearly WT levels of cell surface binding to cells expressing Wnt7a, Fz, and Gpr124. The data support a model in which (1) CC1 and CC4 are the most important CC domains for complex formation [Cho et al., 2017 and Figure 3A of this manuscript show that CC5 is largely dispensable], and (2) optimal activity for signaling also requires the right 3D spacing between these domains. These data have been added to the Results section and are presented in Figure 3—figure supplement 1.

4) Norrin interactions (Figure 1F,G): In the context of the provided data on norrin binding and signaling, the authors should cite and discuss the manuscript by Bang et al., 2018, which may explain the finding that Fzd4CRD-Fzd6 did not signal. Specifically, a flexible linker domain between the CRD and TM1 in Fzd4 plays an important role in Norrin signaling, and inclusion of this linker in the Fz4CRD-Fz6 construct may rescue the lack of β-catenin signaling.

Thank you for pointing out that omission on our part, which we have now corrected. We have added a sentence in the Results section and a sentence in the Discussion section briefly summarizing the Bang et al. results and indicating that they are consistent with what we report in Figure 1F and G.